# Epilepsy in Mitochondrial Diseases—Current State of Knowledge on Aetiology and Treatment

**DOI:** 10.3390/children8070532

**Published:** 2021-06-22

**Authors:** Dorota Wesół-Kucharska, Dariusz Rokicki, Aleksandra Jezela-Stanek

**Affiliations:** 1Department of Pediatrics, Nutrition and Metabolic Diseases, The Children’s Memorial Health Institute, Al. Dzieci Polskich 20, 04730 Warsaw, Poland; d.wesol-kucharska@ipczd.pl (D.W.-K.); d.rokicki@ipczd.pl (D.R.); 2Department of Genetics and Clinical Immunology, National Institute of Tuberculosis and Lung Diseases, 26 Plocka Str, 01138 Warsaw, Poland

**Keywords:** epilepsy, mitochondrial disorders, mtDNA, nDNA, treatment, antiepileptic drugs (AED)

## Abstract

Mitochondrial diseases are a heterogeneous group of diseases resulting from energy deficit and reduced adenosine triphosphate (ATP) production due to impaired oxidative phosphorylation. The manifestation of mitochondrial disease is usually multi-organ. Epilepsy is one of the most common manifestations of diseases resulting from mitochondrial dysfunction, especially in children. The onset of epilepsy is associated with poor prognosis, while its treatment is very challenging, which further adversely affects the course of these disorders. Fortunately, our knowledge of mitochondrial diseases is still growing, which gives hope for patients to improve their condition in the future. The paper presents the pathophysiology, clinical picture and treatment options for epilepsy in patients with mitochondrial disease.

## 1. Introduction

Mitochondria are organelles that are present in almost all cells of the body, which are primarily responsible for producing energy (in the form of ATP) by the process of oxidative phosphorylation (OXPHOS), play a role in the homeostasis of calcium ions, take part in signal transduction between cells by producing reactive oxygen species, and also participate in cell apoptosis [1,2]. Mitochondrial diseases (MDs) represent a clinically and genetically heterogeneous group of diseases with a summary incidence estimated at 1.6:5000 live births, making them the most common diseases among inherited metabolic diseases [2,3]. The diseases may result both from a pathogenic variant in all 37 genes of mitochondrial DNA (mtDNA) and damage to nuclear DNA (nDNA). Currently, almost 400 genes related to MD are identified in nDNA [2,4,5]. The clinical picture of MD is very varied, but typically the highly energetic tissues are affected, including the central and peripheral nervous system, skeletal muscles, sense organs, heart, liver, gastrointestinal tract, or endocrine system [2,4,5]. Central nervous system symptoms—regression in development, delayed psychomotor development, or epilepsy—are dominant features in children with MD. Epilepsy is nonetheless a most clinically significant problem among MD patients since seizures are usually difficult to treat and often deteriorate the patient’s cognitive development, leading to epileptic encephalopathy or a worse prognosis. The incidence rate of epileptic seizures among MD subjects is estimated at 10–40%, but it may even reach 60% in paediatric patients [6,7,8,9,10].

## 2. Pathophysiology of Epilepsy in Mitochondrial Diseases

Epileptic seizure is a sudden and excessive neural discharge resulting from uncontrolled depolarisation of the neural membrane, and its spread is caused by impaired mechanisms regulating this transmission, e.g., balance between inhibitory (γ-aminobutyric acid, GABA) and stimulatory (glutamic and aspartic acid) neurotransmitters. Maintaining membrane polarisation requires high energy input and primarily involves calcium and sodium channels [11]. The pathophysiology of epilepsy in MD is not fully known (Figure 1).

The occurrence of seizures is explained mainly by a deficit of energy, ATP, resulting from impaired oxidative phosphorylation, which is the core of MD. ATP molecules are of key importance for the function of sodium–potassium ATPase (Na+/K+ ATPase), which ensures normal polarisation of the neural membrane and maintenance of resting membrane potential. Impairment in the potential leads to neuronal hyperexcitability, which translates into seizures.

Additionally, ATP deficit suppresses the effect of intermediate inhibitory neurons in the hippocampus, facilitating the spread of excitation in the neuronal network [6,12,13]. Lack of ATP molecules also results in the reduced potential of GABA-ergic inhibitory neurons, which leads to an impaired balance between excitatory and inhibitory neurons and excessive cell excitation [13,14]. Moreover, ATP deficit leads to increased glutamate release (excitatory neurotransmitter) from astrocytes to synaptic space and disturbance of the glutamate–aspartate transporter [1,13,15]. Other hypotheses explain seizures in MD by abnormal haemostasis of calcium ions, abnormal function of ion channels in the neural membrane, or neurotransmitter disorders [15]. A growing amount of data suggest that seizures occurring in MD patients may be caused by excessive amounts of reactive oxygen species (ROS), resulting from abnormal mitochondrial function [1,16,17]. Seizures occurring in the course of mitochondrial diseases may also have a structural background. Severe forms of pyruvate dehydrogenase complex (PDHc) deficit involve damage to the brain structure as early as in the intrauterine stage, which may be the source of seizure [18].

Moreover, the seizures themselves, especially those that are repeated or long-lasting, lead to mitochondrial dysfunction and escalate energy deficit (epileptic seizures use large amounts of energy), which provokes subsequent seizures and makes them difficult to suppress. Such a self-perpetuating cycle may lead to the development of stroke-resistant episodes common in MD, especially in mitochondrial encephalopathy, lactic acidosis, and stroke-like episodes (MELAS) and POLG-related diseases (diseases resulting from damage to DNA polymerase gamma), and the occurrence of such episodes is often related to status epilepticus [8,15]. 

Although the pathomechanism of epilepsy in MD patients mentioned above seems probable, this does not explain why seizures only occur in some MD patients. Additionally, there are changes in mtDNA (especially with variants m.3243A>G and m.8344A>G) and nDNA (e.g., mutations in *POLG*) which predispose to the development of epilepsy. The exact phenotypic manifestation is, however, difficult to predict (see Table 1). The degree of heteroplasmy explains only to some extent the severity of mitochondrial diseases. Moreover, as supported by Tranah et al., the accumulation of a rare genetic disease mutation, e.g., m.3243A>G, manifests as several ageing outcomes, and some diseases of ageing may be attributed to the accumulation of mtDNA damage, leading to differing phenotypes [19]. Additionally, seizures are more common in complex I and/or IV of mitochondrial respiratory chain failures than in complex II and III [9,10,20,21,22]. On the other hand, it was observed that MD, depending on changes in specific genes (e.g., *SURF1, OPA1, PEO1,* m.14709), does not involve epileptic seizures, and the pathomechanism of this phenomenon is not known [8,22,23]. Finally, the role of other genetic variants or polymorphism has to be defined. As discussed by Pickett et al., age, age-adjusted blood heteroplasmy levels, and sex are poor predictors of phenotypic severity. Still, the provided results showed good evidence for the presence of nuclear genetic factors influencing clinical outcomes in m.3234A>G-related disease [24].

## 3. Clinical Picture of Epilepsy in Mitochondrial Diseases

The onset of seizures in MD patients may occur at any age. Seizures may be one of the first symptoms of mitochondrial disease in children (in nearly 20%), but in most patients, they occur as the disease progresses and changes in the central nervous system (CNS) become more severe, e.g., in the course of recurrent stroke-like episodes, or with the progression of other neurodegenerative changes [23]. Typically, epilepsy is one of many other MD symptoms, and it is the most common feature of CNS involvement [8,33]. In certain mitochondrial diseases, seizures are part of a syndrome—e.g., in Alpers–Huttenlocher syndrome (AHS) and other phenotypes associated with a pathogenic variant in the *POLG* gene, in a deficit of pyruvate dehydrogenase complex (PDHc), myoclonic epilepsy with ragged red fibres (MERRF), MELAS, or in Leigh syndrome. A short characterisation of the diseases mentioned above is presented in Table 1 [25,26,27,28,29,30,31,32].

In patients showing epilepsy in the course of MD, the onset of the disease’s symptoms occur much earlier than in subjects without seizures. Patients with MD and epilepsy more often presented perinatal symptoms (e.g., disorders in the intrauterine development or hypertrophic cardiomyopathy) and delayed or impaired development than subjects without epilepsy [8,9,10]. 

MD patients most often experience myoclonus and various types of focal seizures, but the seizure may also have any other morphology: tonic seizures, tonic–clonic seizures, infantile spasms, or even, occasionally, typical absence seizures. From 20–60% of patients experience various types of seizures [4,8,9,10]. Epileptic seizures may form specific epilepsy syndromes—such as West syndrome, Ohtahara syndrome, Lennox–Gastaut syndrome, and Landau–Kleffner syndrome [9,20,23,33]. In the majority of patients (>92%), seizures frequently recur, and in 27%, their occurrence is considered very frequent (every day or every week) [10].

Aside from refractory and frequently recurring seizures, MD patients experience status epilepticus, including nonconvulsive status epilepticus and epilepsia partialis continua (EPC)—a focal motor status epilepticus (spontaneous regular or irregular clonic muscular twitching affecting a limited part of the body, sometimes aggravated by action or sensory stimuli, occurring for a minimum of one hour, and recurring at intervals of no more than ten seconds) [34]. They are difficult to diagnose and treat, thus resulting in a poor prognosis. Status epilepticus in MD is more common in patients with damage to mtDNA (especially in MELAS and MERRF syndromes) and mitochondrial depletion syndromes (especially with pathogenic changes in *POLG*) and are co-existent with stroke-like episodes [1,25,26,32,35]. EPC has been observed in subjects with pathogenic variants in *POLG* and mtDNA. In addition, this may be the first epilepsy manifestation in these subjects. EPC is related to a very poor prognosis—out of 12 paediatric patients with MD and EPC, in only two patients were the seizures partially controlled, in 2/12 disease progression occurred, and 8/12 died within a year of the EPC episode [34]. 

## 4. Diagnostics of Epilepsy in Mitochondrial Disease

### 4.1. Electroencephalography in Patients with Mitochondrial Disease

There is no typical EEG trace for seizures in MD. Background activity in most patients, even without clinical symptoms, is disturbed: 109/165 (66%) of patients had abnormal EEG, mainly in the form of slow background activity with a large proportion of delta waves and lack of spatial differentiation or trace asymmetry [9,21]. Epileptiform changes in EEG of MD subjects typically involve focal changes (23–71%) and multifocal changes (35–56%), sometimes generalised changes (13–25%), or hypsarrhythmia [36,37]. Most frequently, seizures are propagated from the occipital lobe and posterior quadrant of temporal and parietal lobes. Myoclonic seizures involve spikes and polyspikes, which may be activated by photostimulation or opening of the eyes. However, it must be remembered that such seizures, especially in MERRF, are not always caused by epileptic activity, but they may result from cerebellar or medullary dysfunction [15,35]. On the other hand, Alpers–Huttenlocher syndrome typically involves occipital rhythmic high-amplitude delta with superimposed (poly)spikes (RHADS), but this is not a pathognomonic trace for this syndrome only [38]. About 10% of MD patients show normal EEG despite epileptic seizures; however, these are patients with occasional seizures [10].

### 4.2. Neuroimaging 

Neuroimaging changes in MD are quite characteristic for this group of patients, but they are not recognisable. MD′s typical features include bilateral symmetrical signal abnormality in the basal ganglia, brain stem, thalamus, and/or cerebellum hyperintensities in T2 and FLAIR [4,8,21]. White matter may be diffusely abnormal (leukodystrophy) [8,20]. There may also be structural brain abnormalities like agenesis of the corpus callosum, and ventriculomegaly as an effect of energy deficiency during brain formation [29]. A definite lactate peak is observed in MD by proton MR spectroscopy [21]. The analysis of 1467 patients with MD revealed that MRI abnormalities were significantly more common (*p* < 0.001) in subjects with epilepsy than in MD patients without seizures (88 vs. 54%) [10]. Brain atrophy was also more common in the group of epileptic subjects [36,38]. Moreover, patients with MD and epilepsy most often report stroke-like changes, as well as changes in basal ganglia and white matter [33,36]. 

## 5. Pharmacological Treatment of Epilepsy in Patients with Mitochondrial Disease

Treatment of mitochondrial disorders is a challenge for physicians and researchers. Most interventions and guidelines are related to symptomatic treatment, with supplementation of cofactors, vitamins, or antioxidants, and mild exercises are recommended. Despite numerous studies, the efficacy of most of these interventions has not been confirmed [39,40,41,42,43]. 

There is also no established scheme of epilepsy treatment in mitochondrial diseases; therefore, general principles of epilepsy treatment are applied [44]. First-line therapy often includes levetiracetam (LEV), frequently combined with clonazepam (CZP), clobazam (CLB), or topiramate (TPM). Zonisamide (ZNS) is also safe, but there are few literature reports of patients treated with this medication [31,35,45]. Lamotrigine (LTG) may promote myoclonic seizures and has not always been effective in patients with MD. Some experts recommend phenobarbital (PB) or primidone (PRM), but there are few studies on the use of these drugs, and not all of them showed efficacy [35]. There are also reports on perampanel (PER) efficacy in treating status epilepticus in subjects with MELAS [46]. In most patients, seizures are intractable, which often require polytherapy with two or three medications. Only 5–30% of patients had seizures controlled with one antiepileptic drug only. In nearly 8% of patients, it was possible to discontinue treatment [8,33].

Treatment of status epilepticus is also challenging. The management of MD patients is similar to the management of status epilepticus in other patients, except for avoiding the administration of valproic acid. First-line therapy includes benzodiazepines—e.g., midazolam and LEV (20–40 mg/kg, max 4500 mg) intravenous (iv), but the use of phenytoin (15–20 mg/kg), phenobarbital (10–15 mg/kg), or lacosamide (200–400 mg) is also possible. One must bear in mind that MD patients are more susceptible to the development of propofol infusion syndrome. Although propofol is not contraindicated in patients with MD, caution is recommended while using this drug in this group of subjects [15,47,48].

Regarding contraindicated substances, the only absolutely contraindicated drug in treating epilepsy in MD patients is valproic acid (VPA), which is especially relevant for diseases associated with *POLG* pathogenic variants. The drug may induce fulminant hepatic impairment in these subjects [25,49]. Additionally, VPA may cause secondary carnitine deficit, especially in patients with damage to complex I and IV of the mitochondrial respiratory chain [6]. There are single reports of successful VPA treatment of epilepsy in subjects with mitochondrial disease. However, caution is always recommended before introducing this drug in patients with suspected MD [36,39]. Experts suggest VPA in MD patients without a pathogenic variant in *POLG*, and without liver disease, for the treatment of refractory epilepsy [47]. In patients with mitochondrial depletion syndrome, one should avoid the use of vigabatrin (VGB), which inhibits the conversion of deoxyribonucleoside diphosphate (ADP) to deoxyribonucleoside triphosphate (ATP), whereby it increases mtDNA depletion. When using topiramate (TPM), it must be remembered that it potentiates acidosis [39]. On the other hand, the toxicity of carbamazepine (CBZ), phenytoin (PHT), or phenobarbital (PB) outweighs their efficacy, so these agents should be avoided in the treatment of MD patients [50]. A list of safe antiepileptic drugs which may be used in patients with mitochondrial disease is presented in Table 2.

In patients with the m.3243A>G variant, the administration of L-arginine has been confirmed to reduce the incidence of stroke-like episodes and thus reduce the risk of epilepsy and status epilepticus in the course of such episodes [52]. There are also single reports in subjects with epilepsy and Kearns–Sayre syndrome diagnosed with folate (5-methyltetrahydrofolate) deficit, where folic acid supplementation in these subjects was related to improvement in seizure control [53]. Next, high doses of co-enzyme Q10 in patients with primary co-enzyme Q10 deficiency may reduce epileptic seizures in this group of subjects [35].

MD therapy must not exclude such diseases, where it is possible to apply causal treatment or at least mitigation of disease symptoms. Examples of such conditions with specific procedures are presented in Table 3.

## 6. Non-Pharmacological Treatment of Epilepsy in Patients with Mitochondrial Disease

An alternative treatment for epilepsy, aside from standard antiepileptic drugs, is a ketogenic diet (KD). This is a verified and recommended procedure both in adults and children, including infants [60,61,62,63]. There are reports of successful epilepsy treatment with a ketogenic diet in MD patients, especially those with impaired complex I of the mitochondrial respiratory chain [21,64]. A ketogenic diet is based on the supply of small amounts of carbohydrates to the benefit of fats. This leads to the formation of ketone bodies, which represent an alternative source of energy for cells. The antiepileptic mechanism of KD involves a reduction in the glutamate level in the sympathetic space, which reduces neuronal excitation.

Additionally, decanoic acid present in KD is a strong direct receptor inhibitor for glutamine—α-amino-3-hydroxy-5-methyl-4-isoxazolepropionic acid receptor (AMPA)—which directly translates into reduced neuronal excitation. An additional benefit, especially for MD patients, is that ketone bodies are a source of energy [13]. Although the mechanism of action of KD in MD patients is not fully explained, it is known to improve the cell energy profile, lead to the stimulation of mitochondrial biogenesis in skeletal muscles, prevent the formation of abnormal mitochondria, increase ATP production in the respiratory chain, and reduce the number of COX-negative fibres (a marker of mitochondrial damage) in skeletal muscle biopsy [65,66,67].

Numerous studies confirmed KD efficacy in patients with mitochondrial disease (both in adults and children), mainly concerning reducing epileptic seizures. One showed remission of epileptic seizures in 50% of patients (12/24) treated with a KD [21]. Moreover, other studies showed that aside from better seizure control (seizure reduction >50% in 8/20 patients after one year and 7/20 patients after two years of a KD), all patients showed improvement in cognitive functions [68]. A KD may also improve muscular strength and reduce mtDNA heteroplasmy [67].

Other options for epilepsy treatment, especially in the case of refractory epilepsy, are vagal stimulation, deep brain stimulation, or palliative surgical treatment [26]. There are few literature data on the use of these approaches to treat epilepsy in patients with MD. VNS implantation was effective in two patients, resulting in a reduction in seizures of >50% [9]. Palliative neurosurgical treatment was described in 4/40 patients with MD and Lennox–Gastaut syndrome [20].

## 7. Prognosis in Epilepsy in Patients with Mitochondrial Disease

Persistent seizures and status epilepticus lead to neural damage, astrocyte gliosis, damage to myelin, and, as a result, brain atrophy [5]. As confirmed by neuroimaging in subjects with MD and epilepsy, this group more often showed brain atrophy than those with MD and without epilepsy [36,69]. The occurrence of seizures in children is related to a worse prognosis regarding development and survival, especially if epilepsy occured at < 1 year. Seizures are often intractable, and epilepsy leads to progressive neurodegenerative changes and epileptic encephalopathy [6,9,20]. In a group of 56 paediatric patients with MD, 45% (22/56) of patients died, including half of the patients within nine months of the first seizure [33]. In another study, in a group of 46 children with MD and epilepsy, 11 children died within one year of the occurrence of epilepsy [69].

## 8. Conclusions

The mechanism of epilepsy development in mitochondrial diseases is a subject of ongoing studies, while the treatment of epilepsy is challenging for both physicians and scientists. Current studies, primarily involving multi-omic analyses, provide a better understanding of the mechanism leading to the development of such changes, which gives a chance for future detailed diagnostics and knowledge of impaired metabolic pathways, and, most of all, gives hope for the development of individualised treatment of patients [2,70,71].

## Figures and Tables

**Figure 1 children-08-00532-f001:**
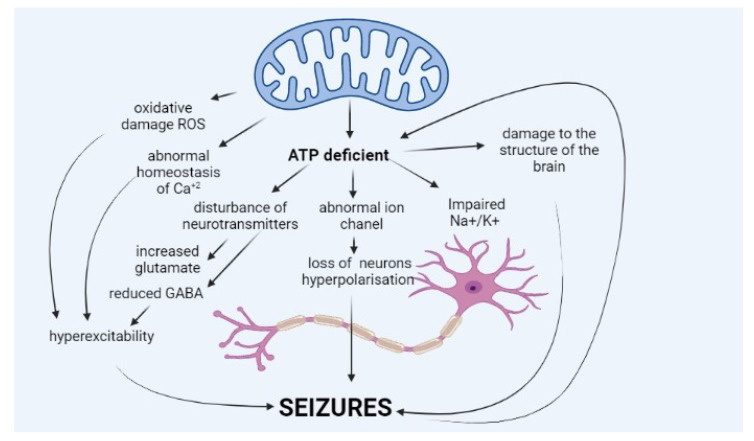
Pathophysiology of epilepsy in mitochondrial diseases. (GABA: γ-aminobutyric acid, Na+/K+: sodium–potassium ATPase, ROS: reactive oxygen species).

**Table 1 children-08-00532-t001:** The most frequent mitochondrial diseases with epilepsy (causative genetic variants are cited from OMIM database, https://www.omim.org and MITOMAP, https://www.mitmap.org/MITOMAP, accessed 2 June 2021).

Disease	Gene	Clinical Picture	Treatment
Alpers–Huttenlocher syndrome (AHS) [25,26]	*POLG* (nDNA)	Progressive neurodegeneration, refractory seizures, movement disorder, neuropathy and hepatic failure, focal-onset seizures predominate, but seizure may also tonic-clonic, or myoclonic; 68% developed status epilepticus and 58% epilepsia partialis continua, status epilepticus is the leading cause of death in children with AHS	In case of refractory seizures, polytherapy is necessary (with no dedicated drug; however, valproic acid is absolutely contraindicated)
Pyruvate dehydrogenase complex deficiency (PDHc) [27,28]	*PDHA, PDHB, LIAS, LIPT1, DLD, PDH,* (nDNA)	Epilepsy begins in infancy with infantile spasms, clonic seizures or refractory focal epilepsy, developmental delay, ataxia, hypotonia, hypertonia, abnormal eye movements, dystonia, axonal neuropathy	The ketogenic diet is the treatment of choice; in some individuals, improvement after thiamine supply possible
Leigh syndrome (LS) [29]	More than 90 genes (nDNA and mtDNA)	Typical features include: (1) developmental regression or developmental delay, (2) specific basal ganglia/brain stem changes bilaterally, and (3) abnormal mitochondrial energy metabolism; epileptic seizures are frequent, both focal and generalised	Due to frequent drug-refractory seizures, polytherapy is often necessary
Myoclonic epilepsy with ragged red fibres (MERRF) [30,31]	The most common pathogenic variants in mtDNA, *MTTL1* (80%): m.8344A>G; *MTTK* (10%): m.8356T>C, m.8363G>A, m.8361G>A	Onset usually in adults, 30% in childhood. Progressive myoclonic epilepsy is part of the phenotype, but seizures can be often generalised tonic, clonic or atonic. Seizure was reported in 33% to 100% of patients; co-occurs with cerebellar ataxia, cardiac arrhythmias, myopathy, diabetes, hearing loss, dementia	The combination of levetiracetam with carbamazepine may have the strongest beneficial effect on myoclonic seizures
Mitochondrial encephalopathy, lactic acidosis, and stroke-like episodes (MELAS) [32]	The most common pathogenic variants in mtDNA, *MTTL1* gen: m.3243A >G m.3271T>C; *MTND5* gen: m.13513G>A	Focal and generalised seizures are possible, preceded by or associated with migraine-like headache; the most typical are seizures in the course of a stroke-like episode, focal status epilepticus with a secondary encephalopathy is common	L-arginine and/or citrulline as prevention and treatment of stroke-like episodes

**Table 2 children-08-00532-t002:** Safety of antiepileptic drugs in mitochondrial diseases.

Mitochondria-Safe AEDs	AEDs to Use Carefully	AEDs Which Could Aggravate Myoclonus
Benzodiazepine [47,51] Gabapentin [47,51] Lacosamide [47,51] Lamotrigine [47,51] Levetiracetam [10,47,51] Oxcarbazepine [10,47,51] Peranpanel [46,47,51] Rufinamide [47,51] Stiripentol [10,47,51] Zonisamide [47,51]	Valproic acid—contraindicated in *POLG* mutations [25,39,51] Vigabatrin—may need to be avoided in patients with mtDNA depletion syndromes [39] Topiramate—may worsen acidosis [39] Phenytoin * [50] Carbamazepine * [50] Phenobarbital * [50]	Valproic acid [35] Phenobarbital [35] Lamotrigine [35] Phenytoin [35] Carbamazepine [35] Oxcarbazepine [35] Vigabatrin [35] Tiagabine [35] Gabapentin [35] Pregabalin [35]

* Toxic effect on mitochondria outweighs the beneficial effect.

**Table 3 children-08-00532-t003:** Currently available treatment options in mitochondrial diseases.

Disease (Gene)	Clinical Features	Treatment
Primary co-enzyme Q_10_ deficiency (*COQ2, COQ4, COQ5, COQ6, COQ7, COQ9, PDSS1, PDSS2*)	Multisystem involvement with progressive neurological dysfunction, seizures, encephalopathy, stroke-like episodes, cerebellar ataxia, pyramidal dysfunction, cognitive impairment renal failure, and steroid-resistant nephrotic syndrome	High-dose oral CoQ_10_ supplementation (ranging from 5 to 50 mg/kg/day) [54]
Pyruvate dehydrogenase complex (PDHc) deficiency (*PDHA1, PDHB, LIAS, PDP1, PDHX, DLAT*)	Epilepsy, developmental delay, ataxia, hypotonia, hypertonia, abnormal eye movements, dystonia, ataxia, axonal neuropathy, and poor feeding	Ketogenic diet 3:1–4:1 and thiamine (50 mg/kg/day, max 300–900 mg/day) [18,27]
ACAD9 deficiency (*ACAD9*)	Hypertrophic cardiomyopathy, lactic acidosis, exercise intolerance, and occasional seizures	Riboflavin (vitamin B2) 20 mg/kg/day–max 400 mg/day [55]
Impairment of thiamine transport and metabolism (*SLC19A3, SLC19A2, SLC25A19*, *TPK1*)	Biotin–thiamine-responsive basal ganglia disease or Leigh syndrome; subacute encephalopathy with confusion, dysphagia, dysarthria, seizures, external ophthalmoplegia, and generalised stiffness following a history of febrile illness; progresses to severe quadriparesis, rigidity, dystonia, coma, and death if early treatment is not administered	Biotin (5–10 mg/kg/day) and thiamine (10–40 mg/kg/day, between 300 and 900 mg/day) [56]
AGC1 deficiency (aspartate–glutamate carrier isoform 1) (*SLC25A12*)	Severe hypotonia, arrested psychomotor development, and seizures from a few months of age, a global lack of myelination in the cerebral hemispheres	Ketogenic diet 3:1–4:1 [57]
Ethylmalonic encephalopathy (*ETHE1*)	Early onset, progressive disorder, developmental delay, generalised infantile hypotonia that evolves into hypertonia, spasticity and dystonia; generalised tonic–clonic seizures; and generalised microvascular damage	N-acetylcysteine in combination with metronidazole [58]
Beta-hydroxyisobutyryl-CoA deacylase deficiency (*HIBCHD*)	Progressive neurodegenerative disorder, associated with basal ganglia changes on brain magnetic resonance imaging; elevated hydroxy-C4-carnitine levels	Low-valine and high-carbohydrate diets, antioxidants (co-enzyme Q10, vitamin E, vitamin C), carnitine, and N-acetylcysteine [59]

## Data Availability

Not applicable.

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
