# Peer review of "Epilepsy in Mitochondrial Diseases—Current State of Knowledge on Aetiology and Treatment"

_children, 2021, doi:10.3390/children8070532_

Round 1
Reviewer 1 Report
Manuscript is well organized and clearly written. Authors show exhaustive picture of pathophysiology of epilepsy in mitochondrial diseases and its management. I’d like only to point out the abbreviation of ketogenic diet (KD), precisely cited at the start of paragraph but several times wrongly referred at page 7. Minor spell check required.
Author Response
Dear Reviewer,
we’d like to thank you very much for your input, constructive comments and valuable suggestions.
The manuscript has been revised accordingly, and answers to all the notes have been provided, point-by-point, and described below.
We do hope now, that in the improved version our paper meets with your positive feedback.
Reviewer 1
Comments and Suggestions for Authors
Manuscript is well organized and clearly written. Authors show exhaustive picture of pathophysiology of epilepsy in mitochondrial diseases and its management. I’d like only to point out the abbreviation of ketogenic diet (KD), precisely cited at the start of paragraph but several times wrongly referred at page 7. Minor spell check required.
thank you for pointing this out -> the wrong abbreviation has been fixed
Reviewer 2 Report
The review article by Wesol-Kucharska et al. is well organized, balanced in its parts, and critically discusses all relevant issues related to epilepsy in MD.
I have only two minor suggestions regarding the reference of Table 1 and 2: it is not sufficient to mention the reference in the table legend. The Authors should include this information in the table as they did in Table 3. This will make it easier and more direct for the reader to relate the information to the original writing. I would also recommend avoiding mentioning reviews and using research articles instead.
Before publication, the article should be linguistically revised by an English-speaking colleague or an editor.
Author Response
Dear Reviewer,
we’d like to thank you very much for your input, constructive comments and valuable suggestions.
The manuscript has been revised accordingly, and answers to all the notes have been provided, point-by-point, and described below.
We do hope now, that in the improved version our paper meets with your positive feedback.
Reviewer 2
Comments and Suggestions for Authors
The review article by Wesol-Kucharska et al. is well organized, balanced in its parts, and critically discusses all relevant issues related to epilepsy in MD.
I have only two minor suggestions regarding the reference of Table 1 and 2: it is not sufficient to mention the reference in the table legend. The Authors should include this information in the table as they did in Table 3. This will make it easier and more direct for the reader to relate the information to the original writing. I would also recommend avoiding mentioning reviews and using research articles instead.
thank you very much, indeed we’ve missed it -> the reference in Table 1 and 2 have now been added, as well as 3 references has been omitted
Before publication, the article should be linguistically revised by an English-speaking colleague or an editor.
thank you! the paper, before submission, underwent linguistic check but – according to this suggestion – we’ve corrected yet

Reviewer 3 Report
This paper review the pathophysiology, clinical picture and treatment options for epilepsy in patients with mitochondrial disease. Interesting and can be relevant for everyday clinical practice, however there are no clear novel findings, need thorough revision based on recent publications.
Abstract:
The paper is on mitochondrial epilepsy. I recommend focusing on mitochondrial epilepsy rather than mitochondrial medicine in general.
1- Introduction:
Please see my comment over. I would prefer to focus on epilepsy rather than on mitochondrial medicine in general.
- 25: occurring in almost every: suggest change to present in almost.
- 26: in the process of: suggest change to by the process of.
- 31: 1:5000 live births: recent paper showed a figure of 1:4200, please have another look at the literature.
- 32: an abnormal variant in: suggest pathogenic variant
- Please check the English language throughout the manuscript.
2- Pathophysiology of epilepsy in mitochondrial diseases
It could be a good idea to summarize all these hypothesis in a figure so it will be easier to follow. I agree there are many theories and no clear answer.
There are no novel findings here and these theories have been described in several previous publications. Could the authors link the pathophysiology with the phenotype? Phenotype related to occipital lobes? Different pathophysiology mtDNA versus nDNA ? age of onset ? severity ?
3- Clinical picture of epilepsy in mitochondrial diseases
90- (in nearly 20%) : in adults or pediatric ? please cite a reference here.
96- other POLG-pathies: what the authors mean by other POLG pathies ?
Table 1: the table missing many other pathogenic mutations both in mtDNA and nDNA which are associated with epilepsy.
Table 1 . Treatment:
POLG : the authors did not mention which treatment they recommend ?
MERFF: why the author recommend these AMES, based on what ?
101-102 : In patients showing epilepsy in the course of MD, the onset of disease’s symptoms occurred much earlier than in subject without seizures : please cite , not totally agree.
109-111 : Epileptic seizures may form specific epilepsy syndromes – such as West syndrome, Ohtahara syndrome, Lennox–Gastaut syndrome, Landau–Kleffner syndrome. In the majority of patients (> 92%), seizures frequently recur, and in 27% their 111 occurrence is considered very frequent (every day or every week): I do not think this is correct. Please remove, or cite each of these syndrome with ref that link to mutation in mtDNA eller nDNA.
- Diagnostics of epilepsy in mitochondrial disease
145 : Neuroimaging: Could the author summarize the neuro-imaging, MRI changes ? T2, DWI etc.
146 : Neuroimaging changes in MD are quite characteristic for this group of patients: could the authors cite a ref.
- Pharmacological treatment of epilepsy in patients with mitochondrial disease
182-186: There are single reports of successful VPA treatment of epilepsy in subjects with mitochondrial disease. 183 However, caution is always recommended before introducing this drug in patients with suspected MD [33,37]. Experts allow the use of VPA in MD patients without a pathogenic 185 variant in POLG, and without liver disease, for the treatment of refractory epilepsy.
I suggest remove this. VPA is contraindicated in patients with suspect mitochondrial disorders.
- Non-pharmacological treatment of epilepsy in patients with mitochondrial disease
Could the other comment on other non-pharmacological approach as VNS.
Author Response
Dear Reviewer,
we’d like to thank you very much for your input, constructive comments and very valuable suggestions.
The manuscript has been revised accordingly, and answers to all the notes have been provided, point-by-point, and described below.
We do hope now, that in the improved version our paper meets with your positive feedback.
We’re really grateful for you review!
Reviewer 3
Comments and Suggestions for Authors
This paper review the pathophysiology, clinical picture and treatment options for epilepsy in patients with mitochondrial disease. Interesting and can be relevant for everyday clinical practice, however there are no clear novel findings, need thorough revision based on recent publications.
Abstract:
The paper is on mitochondrial epilepsy. I recommend focusing on mitochondrial epilepsy rather than mitochondrial medicine in general.
1- Introduction:
Please see my comment over. I would prefer to focus on epilepsy rather than on mitochondrial medicine in general.
thank you very much for this suggestion. We do agree that the aim of the paper is to present epilepsy related to mitochondrial diseases.
We have decided to make this brief and general introduction with a view to physicians who are not familiar enough with this disease, e.g., those who may not see MD patients on a daily basis but only have occasional contact with an epilepsy patient. As an introduction, if this is ultimately accepted by the reviewer, we would ask that the fragment be left.
- 25: occurring in almost every: suggest change to present in almost.
thank you very much for this suggestion -> it has been corrected
- 26: in the process of: suggest change to by the process of.
thank you very much also for this suggestion -> it has been corrected
- 31: 1:5000 live births: recent paper showed a figure of 1:4200, please have another look at the literature.
thank you -> the data has been corrected and relevant publication added
.- 32: an abnormal variant in: suggest pathogenic variant
thank you very much for this suggestion -> it has been corrected
- Please check the English language throughout the manuscript.
thank you! the paper, before submission, underwent linguistic check but – according to this suggestion – we’ve corrected yet
2- Pathophysiology of epilepsy in mitochondrial diseases
It could be a good idea to summarize all these hypothesis in a figure so it will be easier to follow. I agree there are many theories and no clear answer.
thank you for this excellent suggestion -> Figure has now been submitted
There are no novel findings here and these theories have been described in several previous publications. Could the authors link the pathophysiology with the phenotype? Phenotype related to occipital lobes? Different pathophysiology mtDNA versus nDNA ? age of onset ? severity ?
thank you very much
The reviewer is absolutely right and such a remark is justified. However, our aim was to present the current state of knowledge and that is why we referred to previous publications in this paragraph. Therefore, for an expert in mitochondrial diseases (which the reviewer undoubtedly is), there is no novelty here.
As suggested, we’ve added new references and discus in more details the possible pathomechanisms related to MD clinical diversity.
3- Clinical picture of epilepsy in mitochondrial diseases
90- (in nearly 20%) : in adults or pediatric ? please cite a reference here.
thank you, it was our undersign, the data refer to children -> the reference has been added
96- other POLG-pathies: what the authors mean by other POLG pathies?
thank you for the notice the phrase has been corrected now
Table 1: the table missing many other pathogenic mutations both in mtDNA and nDNA which are associated with epilepsy.
yes, we absolutely agree and thank you for this comment [there are over 200 phenotypes of mitochondrial diseases, thus in the Table we focus only on the most frequent] -> the title has been modified and source were added
Table 1. Treatment: POLG: the authors did not mention which treatment they recommend ?
thank you for this comment. We have mentioned it, but now the sentences are rephrased (both in text and in Table 1) to underline VPA importance/contraindication for POLG individuals
MERFF: why the author recommend these AMES, based on what ?
thank you for these remark and concern -> we have referred to 26 and 32 (is now added in Table 1)
101-102 : In patients showing epilepsy in the course of MD, the onset of disease’s symptoms occurred much earlier than in subject without seizures : please cite , not totally agree.
thank you very much for this comment we have referred to:
[8] Matricardi, S.; Canafoglia, L.; Ardissone, A.; et al. Epileptic phenotypes in children with early-onset mitochondrial diseases. Acta Neurol. Scand. 2019, 140(3), 184-193.
“Retrospective observational study on a cohort of 129 children with mitochondrial disorders with CNS involvement and disease onset during the first year of life. Seizures occurred in 48% (n=62), and the presence of epilepsy was significantly associated with earlier age at disease onset (3 mo [1‐5] 5 mo [2‐9]), presence of perinatal manifestations (35/62 (56.45%) vs14/67 (20.90%)), and early detection of developmental delay and regression (45/62 (72.58%) 29/67 (43.28%)) (P < 0.001).”
[10] Ticci, C.; Sicca, F,; Ardissone, A.; et al. Mitochondrial epilepsy: a cross-sectional nationwide Italian survey. Neurogenetics 2020, 21(2), 87-96 „Among the 1467 MD patients (mean age at MD onset 23.4 ± 19.5 years) 10.0%(147/1467) were reported as suffering from epilepsy. Age at MD onset was lower in the “epilepsy” group (P < 0.001)”
109-: Epileptic seizures may form specific epilepsy syndromes – such as West syndrome, Ohtahara syndrome, Lennox–Gastaut syndrome, Landau–Kleffner syndrome. In the majority of patients (> 92%), seizures frequently recur, and in 27% their 111 occurrence is considered very frequent (every day or every week): I do not think this is correct. Please remove, or cite each of these syndrome with ref that link to mutation in mtDNA eller nDNA.
thank you very much for your careful reading -> the references have been added, as follow:
Epileptic seizures may form specific epilepsy syndromes – such as West syndrome, Ohtahara syndrome, Lennox–Gastaut syndrome, Landau–Kleffner syndrome - were reported in publications in patients with MD:
[9]. Khurana, D.S.; Salganicoff, L.; Melvin, J.J.; et al. Epilepsy and respiratory chain defects in children with mitochondrial encephalopathies. Neuropediatrics. 2008, 39(1), 8-13.
[19]. Lee, S.; Baek, M.S.; Lee, Y.M. Lennox-Gastaut Syndrome in Mitochondrial Disease. Yonsei. Med. J. 2019, 60(1), 106-114.
[23]. El Sabbagh, S.; Lebre, A.S.; Bahi-Buisson, N.; et al. Epileptic phenotypes in children with respiratory chain disorders. Epilepsia 2010, 51(7), 1225-1235.
“In the majority of patients (> 92%), seizures frequently recur, and in 27% their occurrence is considered very frequent (every day or every week)” - these data are from:
[10] Ticci, C.; Sicca, F,; Ardissone, A.; et al. Mitochondrial epilepsy: a cross-sectional nationwide Italian survey. Neurogenetics2020, 21(2), 87-96.
also Lee S et al. mentions that the mean seizure frequency in LS patients was 7.1 per month (1-24), so it is every week too [34].
[34] Lee, S.; Na, J.H.; Lee, Y.M. Epilepsy in Leigh Syndrome With Mitochondrial DNA Mutations. Front. Neurol. 2019, 10, 496.
- Diagnostics of epilepsy in mitochondrial disease
145 : Neuroimaging: Could the author summarize the neuro-imaging, MRI changes ? T2, DWI etc.
thank you for this valuable suggestion description has been added
146 : Neuroimaging changes in MD are quite characteristic for this group of patients: could the authors cite a ref.
thank you for this comment the reference has been added
- Pharmacological treatment of epilepsy in patients with mitochondrial disease
182-186: There are single reports of successful VPA treatment of epilepsy in subjects with mitochondrial disease. 183 However, caution is always recommended before introducing this drug in patients with suspected MD [33,37]. Experts allow the use of VPA in MD patients without a pathogenic 185 variant in POLG, and without liver disease, for the treatment of refractory epilepsy.
I suggest remove this. VPA is contraindicated in patients with suspect mitochondrial disorders.
thank you for this important remark we recommended the avoidance of VPA in the treatment of MD patients (as clearly stated in the first sentence of the mentioned paragraph). However, according to the last recommendations of the group of experts (according to [46]) “In non-POLG patients with mitochondrial disease, without liver disease, valproic acid could be used to manage refractory epilepsy and refractory mood disorders ". Thus, we would like to have this information in the text, but depending on the Reviewer’s opinion (and we appreciate Her/His understanding).
- Non-pharmacological treatment of epilepsy in patients with mitochondrial disease
Could the other comment on other non-pharmacological approach as VNS.
thank you for the suggestion description has now been added

Round 2
Reviewer 3 Report
I have no further comments.